# Load Estimation of Moving Passenger Cars Using Inductive-Loop Technology

**DOI:** 10.3390/s23042063

**Published:** 2023-02-12

**Authors:** Zbigniew Marszalek, Krzysztof Duda, Piotr Piwowar, Marek Stencel, Tadeusz Zeglen, Jacek Izydorczyk

**Affiliations:** 1Department of Measurement and Electronics, AGH University of Science and Technology, 30 Mickiewicz Avenue, 30-059 Krakow, Poland; 2Department of Telecommunications and Teleinformatic, Silesian University of Technology, 16 Akademicka, 44-100 Gliwice, Poland

**Keywords:** weight-in-motion (WIM), slim inductive-loop, multi-frequency impedance, vehicle magnetic profile (VMP), car load estimation

## Abstract

Due to their lack of driving controllability, overweight vehicles are a big threat to road safety. The proposed method for a moving passenger car load estimation is capable of detecting an overweight vehicle, and thus it finds its application in road safety improvement. The weight of a car’s load entering or leaving a considered zone, e.g., industrial facility, a state, etc., is also of concern in many applications, e.g., surveillance. Dedicated vehicle weight-in-motion measurement systems generally use expensive load sensors that also require deep intervention in the road while being installed and also are calibrated only for heavy trucks. In this paper, a vehicle magnetic profile (VMP) is used for defining a load parameter proportional to the passenger vehicle load. The usefulness of the proposed load parameter is experimentally demonstrated in field tests. The sensitivity of the VMP to the load change results from the fact that the higher load decreases the vehicle clearance value which in turn increases the VMP. It is also shown that a slim inductive-loop sensors allows the building of a load estimation system, with a maximum error around 30 kg, which allows approximate determination of the number of passengers in the car. The presented proof of concept extends the functionality of inductive loops, already installed in the road, for acquiring other traffic parameters, e.g., moving vehicle axle-to-axle distance measurement, to road safety and surveillance related applications.

## 1. Introduction

A passenger car being overweight occurs when a total load exceeds the permissible limit, and it can be caused by the cargo and/or the number of passengers [1]. This is a severe problem because overloaded vehicles pose a significant danger to road traffic [2,3], among others, due to the lack of controllability when braking. For this reason, an overloaded passenger vehicle should be prevented from being on the road. Moreover, an excessive load can be caused by smuggling people or goods; thus, the proposed method could also find its application in surveillance and safety systems.

Current research on vehicle weighing in motion (WIM) systems is mostly concentrated on heavy trucks, e.g., [4,5,6,7], because they significantly damage the road surface. These vehicles are next driven to certify static weigh stations [8]. The result of static weighing may be the basis for imposing a penalty for exceeding the total weight. Efforts to achieve constant and high accuracy of vehicle WIM systems are ongoing. Achieving high accuracy is a prerequisite for using WIM systems in direct enforcement mode [9]. In the considered case of passenger cars, road deterioration is of marginal concern, and we concentrate on road safety.

Known vehicle WIM systems are mostly based on wheel load sensors [10,11,12,13]. Piezoelectric, quartz, fiber optic, and strain gauge load cell sensors are used. The WIM system installation requires a lot of intervention in the pavement, much more than in the case of inductive-loop (IL) sensor technology [14,15]. IL sensors cooperate with wheel load sensors in WIM systems, where they are used to detect the vehicle body, enabling the correct operation of the entire WIM system. So far, IL sensor technology alone has not been tested for possible application in vehicle load estimation. The proposed method uses only the existing IL sensors, and no additional sensors are required; nevertheless, we show that the preselection of overloaded cars can still be obtained.

The proposed measurement system exploits the multi-frequency impedance measurement (MFIM) method widely used in biomeasurements, spectroscopy, and impedance tomography [16,17,18,19,20,21,22]. MFIM has also been used to detect and size DNA fragments [23]. However, as recent research shows, the MFIM system for IL sensors provides more reliable signals for various purposes [24,25]. The MFIM system output signal is called vehicle magnetic profile (VMP).

The hypotheses raised in this work are as follows. It is possible to apply an adaptive IL sensors technology and MFIM system for load estimation of passenger vehicle. The main contribution of this work is the new application of IL sensors to estimate the mass of a passenger car load. The main aim is to experimentally demonstrate the sensitivity of a new load parameter, defined based on VMP, to the vehicle load. This relation originates from the dependence of the passenger vehicle clearance value on the vehicle load mass and finally the VMP. It is also highlighted that the slim IL sensors [24,25] are better suited for the task than the wide IL sensors. They allow the building of a system for estimating the load mass of a passenger vehicle within a maximum error of around 30 kg and approximate estimation of the number of passengers under the assumption of an average passenger mass, e.g., 70 kg. The paper describes the whole IL sensor-based system dedicated to various road traffic measurements that is currently working on the UST-AGH campus. The measurements are taken at different frequencies for increased immunity to electromagnetic interferences. Opposite to existing WIM systems that are used for detecting overloaded heavy trucks that damage the road, we consider the case of overloaded passenger cars, which do not damage the road but still pose a threat for road safety.

## 2. Measurement System

The diagram of the measurement system is shown in Figure 1. The suspension of a passenger car is flexible. The load of the car affects the ground clearance. A loaded car has a lower ground clearance than an unloaded one.

A set of four IL sensors (IL1–IL4) have been installed in the lane of the road where the vehicles pass.

The slim IL sensor is made in the same technology as the wide one. The longevity of IL sensors reaches the life of the pavement in which they are installed and can roughly be estimated for 10–20 years.

The precision of the installation of the slim IL sensor with dimensions of 0.1 m by 3.2 m is estimated at 5 mm and is limited by guiding the saw cutting a groove in the road surface for a wire. Such precision is satisfactory. We have two slim IL sensors on the testbed. No significant differences were observed in the obtained VMPs for both slim IL sensors.

Strict comparison of a slim IL sensor with other WIM load sensor technology is difficult because different sensors generate different output signals. A significant problem of the load sensor is the non-uniform sensitivity over the sensor length. In the case of the applied slim IL sensor, this problem can be neglected thanks to sufficient installation precision.

Each IL sensor has its own impedance parameter measurement channel. Impedance measurement is carried out using the auto-balancing bridge method (ABB). The system includes four processing channels dedicated to four IL sensors. Digital processing is carried out using an industrial NI-PXI computer with a data acquisition card containing an FPGA module. Detailed technical information about the IL sensors construction and the implementation of the MFIM method is given in [24]. Compared to [24], the following significant changes have been additionally applied. Currently we use a new set of excitation frequencies, listed in Table 1, for obtaining R-VMP and X-VMP.

Additionally, in order to improve the quality of the acquired VMPs, signal processing tools such as wavelets [26] and an additional low-pass flat-top digital filter [27] with a bandwidth matched to the vehicle velocity are applied. Examples showing the importance of such filtration in VMP processing are provided in Appendix A in Figure A1, Figure A2 and Figure A3.

Car engines generate electromagnetic interferences (EMIs). In the proposed measurement system, EMIs are detected by applying a notch filter for all excitation frequencies (Table 1) and evaluating the level of the remaining output voltage, which is then recorded. The presence of a high level EMIs, especially with frequencies close to the excitation frequency, disturbs the VMP’s shape. By setting different values of excitation frequencies, we increase the overall robustness of the measurement system against EMIs.

The impedance signal contains a DC component equal in value to the nominal resistance (R) and reactance (X). However, the VMPs have this offset removed. VMP extraction consists of thresholding vehicle presence detection and offset subtraction based on pre-trigger and post-trigger VMP values.

The ADCs in the system have a mode of synchronous sampling. Obtained VMPs and EMIs from individual channels are synchronized in time. Exemplary VMPs and EMIs in the time domain are shown in Figure 2.

The distances between the same size IL sensors on the lane equal 1.5 m. This allows us to use X-VMP with IL1 and X-VMP with IL3 to measure vehicle speed. The velocity is next used to scale the time vector of the VMP samples to the vector of the distance traveled by the vehicle in that time. Then, the VMPs can be shifted into the distance domain by the value of the distance between the Ils. For example, the VMPs derived from IL3 may be shifted in the distance domain and thus superimposed on the VMPs from IL1. The IL sensors are not in the same place in the lane, but the VMP can be rendered as if the IL sensors were on the top of each other in the same place in the lane. By resampling data in the distance-domain VMPs, different data realizations for the same vehicle model passing the measurement testbed with different velocities can be compared.

Exemplary VMPs of a Hyundai ix35 presented as a function of the distance traveled via IL sensors are shown in Figure 3. These VMPs apply to the Hyundai ix35 in unloaded (thick lines) and heavy load (thin dashed lines) conditions. The influence of the load on the extreme values in individual VMPs is clearly visible.

Different cars have different VMPs. However, the same vehicle models have a very similar VMPs especially when compared in the distance domain. The differences in the extreme values depend on the load, which is the main finding of this work. In order to detect the vehicles involved in the experiment, a database of reference VMPs was created for them. 

For automatic vehicle model detection (see Figure 1) the VMPs are normalized up to 1 (for R-VMP) or down to −1 (for X-VMP). Next, the distances between the investigated and the reference VMPs are computed, e.g., the sum of squared differences. The minimum distance clearly indicates the greatest similarity to a given reference vehicle.

The software of the measurement system is hybrid; depending on the task, it uses such programming languages as LabVIEW, Matlab, Python, C, and Bash.

The system can run continuously. The longest operating system time without a reboot is up to a month.

## 3. Experiment

The main aim of the experiment is the verification of the posed hypothesis that there is a relation between the load of selected vehicles and their VMPs, which can further be used for the estimation of the load based on the acquired VMP.

The measurement experiment was carried out on the UST-AGH campus, where a testbed is located on the internal road. On a straight stretch of a road, in one lane, four IL sensors with the outline shown in Figure 1 are installed. The testbed is not equipped with any sensors for the dynamic weighing of vehicles in motion.

Three passenger vehicles were used to conduct the experiment: Mercedes GLA200, Hyundai i30, and Hyundai ix35. In the experiment, each vehicle passed through IL sensors at a velocity between 30 km/h and 50 km/h. The driver tried to maintain a constant velocity while passing through the testbed with IL sensors. Every vehicle made five trips; six passes through the testbed per trip, including three passes in the main direction and three in the return direction.

In the first series, each of the tested vehicles was loaded only with the weight of the driver. In the second series, the vehicles were loaded with the mass of the first passenger, 73 kg, sitting in the front seat next to the driver. In the third series, the vehicles were loaded with an additional weight of 76 kg, in the form of cast iron balls distributed evenly in the space of the passengers’ legs. The total weight of the load was 149 kg. In the fourth series, another mass of 76 kg was added. The total weight of the load was then 225 kg. In the fifth series, another mass of 81 kg was added. The total weight of the load was then 306 kg.

The technical data of the test cars and the weight of the driver along with the percent of coverage of the permissible load (M) are summarized in Table 2.

In total, in the experiment, every vehicle made a total of 30 trips with different loads in individual series. Each trip data was documented. The output voltages in the ABB system, car photo (see Figure A4), and passing time were recorded, and EMIs and VMPs were calculated.

## 4. Results

Each of the 90 trips provided VMPs from which the load parameter was calculated:Load parameter = mean(abs(min(mute(X-VMP, EMI))))(1)
where the mean function calculates the average with six values; the abs function calculates the absolute value; the min function finds minimal value; the mute function mutes these values in X-VMP samples for which the EMI level exceeds 1.5 mV and returns the muted X-VMP signals. The load estimation algorithm only takes X-VMPs and EMIs for slim IL sensors. For example, the operation of the mute function is shown in Figure 4 for disturbed single X-VMP.

Figure 5 shows the reference load as a function of the calculated load parameter for three passenger cars listed in the legend.

Based on the least squares fitted linear function, three sets of scaling factors for the load measurement system were determined. The scaling factors and sensitivities are listed in Table 3.

The sensitivity of the slim IL sensor to the load mass that affects the clearance ranges from 30 to 150 micro ohms per kilogram for tested cars.

The recorded VMPs were also used to verify load measurement errors. The weight of the load in kg was calculated for each trip. The error was defined as the difference between the measurement result and the reference value. The errors are presented in the form of a boxplot [28] in Figure 6. The maximum load measurement error does not exceed 30 kg. Detailed results are summarized in Table A1, Table A2 and Table A3 in Appendix B.

By incorporating the mass of the driver into the mass of the vehicle, which must be driven through the testbed, the results for various experiments can be presented in a clear way. The driver’s weight and the mass of fuel, as well as the mass of additional car equipment, e.g., child seats, spare wheal, fire distinguisher, etc., are taken into account in the offset values in the target measurement system.

In Appendix C, we present the results of the load measurement of another passenger vehicle for which a simplified two-point calibration was carried out.

The results of the load estimation were also checked for the Hyundai i30 traveling without additional load on other days. The weather conditions were generally different than on the day of the main experiment. The results of the load estimation were checked. The maximum error did not exceed 30 kg.

## 5. Discussion

Conducted experiments show that VMPs from the slim IL sensors are suitable for vehicle mass estimation. VMPs from the standard IL sensors, having dimensions 1 m by 2 m, give neither good nor clear results. The off-center passage of the car through the standard IL sensors has a large influence on the error, whereas it is negligible for slim IL sensors. Appendix D covers this issue in more detail.

The proposed method uses existing testbed without any modification. Only the software extension is required. The method is sensitive to the kind of a car suspension and may also be sensitive to suspension malfunction. The method is best suited to comparison of the same vehicle with different loads.

The proposed load parameter is defined with the minimum value and mean value, and it is computationally simple. More computationally advanced load parameters were also investigated, e.g., involving integrals of VMPs, but obtained results turned out to be inferior and thus are not presented.

As the proposed method is dedicated to a moving car, there is no way for measuring the car weight without a driver, as in the static measurements. In the conducted experiment each test car was driven by a different person with a different weight (Table 2). The driver’s weight can be taken into account in the calibration process. The characteristics in Figure 5 will then have an offset equal to the driver’s weight. We also note that a typical gas tank volume is above 50 L, what also influences overall car mass.

The measurement method is sensitive to snow and ice covering the road. For a few cm of a frozen snow and ice layer, it was observed in measurements that the weight of the vehicle could be underestimated by tens of kg. 

The main experiment was conducted in cold, sunny weather in late autumn. However, the weather conditions, similar to temperature, are not expected to influence significantly neither car suspension nor IL sensors performance, and thus the proposed load parameter is considered to be robust against environmental conditions.

The average weight of an adult human in Europe is 70.8 kg [29]. The mass of 50 L of gasoline is 37.5 kg. In the proposed method, the largest error in mass measurement reached 30 kg, which added to the weight of a half-full tank of gasoline equals 48.75 kg. Diesel fuel has a higher density; therefore, 50 L of this fuel weighs 42.5 kg. The weight of a half-full tank of oil added to the maximum measurement error equals 51.25 kg. Considering the above, it is safe to assume that the combined error caused by the method itself and an unknown tank volume does not exceed the average weight of an adult. This means that the estimation of the number of passengers in the car could be reliable. We also note that an average adult weight varies across the world and between genders, e.g., in the USA an average male and female weight equals 90.6 kg and 77.5 kg, respectively [30]. Considering also possible child passengers, some advanced classification methods should be applied for reliable detection of the number of passengers in the car.

The proposed load measurement method can be used in conjunction with the standard WIM system as a part of road traffic data fusion system. Commercial application must be preceded by extensive quantitative testing including diverse vehicles and loads.

## 6. Conclusions

The paper shows that the inductive-loop technology can be successfully used for the estimation of the load of a moving passenger car, which was explained and verified by a field experiment. This is an additional feature for the existing system that extends its capabilities. In summary the inductive-loop sensors technology:Enables obtaining multi-frequency VMPs representing changes in the IL sensor impedance component;Allows the capture of the EMI of a car drive;Allows the use of EMI to indicate and mute disturbances in the VMP;Allows for reliable identification of the car model;For cars where the clearance depends on the load, slim IL sensors enables rough estimation of the load.

Estimated load can be further interpreted as a number of passengers in the car under assumption of the average person’s weight. For a given car, the method can be used for comparing the load of a car when entering and leaving distinguished zone, e.g., city or country border.

We assume that similar results could be obtained for other types of vehicles, e.g., a van, a pickup, truck, etc., but this still has to be tested. We also expect that the load of the electric cars could be estimated by the proposed method, as it is sensitive to the suspension and not the engine, but it still remains to be experimentally investigated.

Existing WIM system are calibrated for heavy trucks and are not used for passenger cars as is the proposed method.

## Figures and Tables

**Figure 1 sensors-23-02063-f001:**
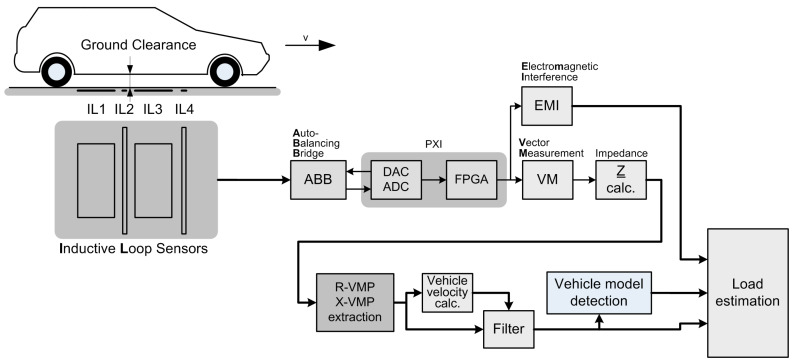
Load estimation system block diagram. IL1—the first standard IL sensor, IL2—the first slim IL sensor, IL3—the second standard IL sensor, IL4—the second slim IL sensor.

**Figure 2 sensors-23-02063-f002:**
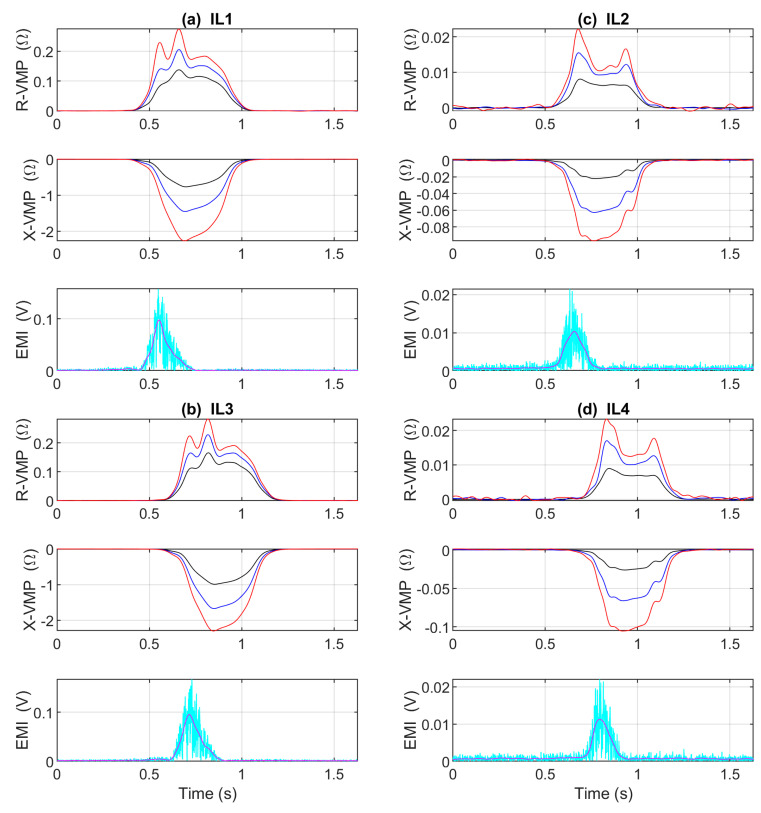
Exemplary VMPs and EMIs of Hyundai ix35 presented as a function of time. R-VMP and X-VMP denote the real and the imaginary impedance components, EMI denotes the electromagnetic interference; (**a**) the first standard IL1 sensor; (**b**) the second standard IL3 sensor; (**c**) the first slim IL2 sensor; and (**d**) the second slim IL4 sensor; red VMPs are obtained at a high frequency component; blue VMPs are obtained at an intermediate value of the frequency component; black at the lowest frequency component, according to the operating frequencies listed in Table 1. The cyan EMI signal is the absolute voltage, the magenta is the average of the low-pass filter output.

**Figure 3 sensors-23-02063-f003:**
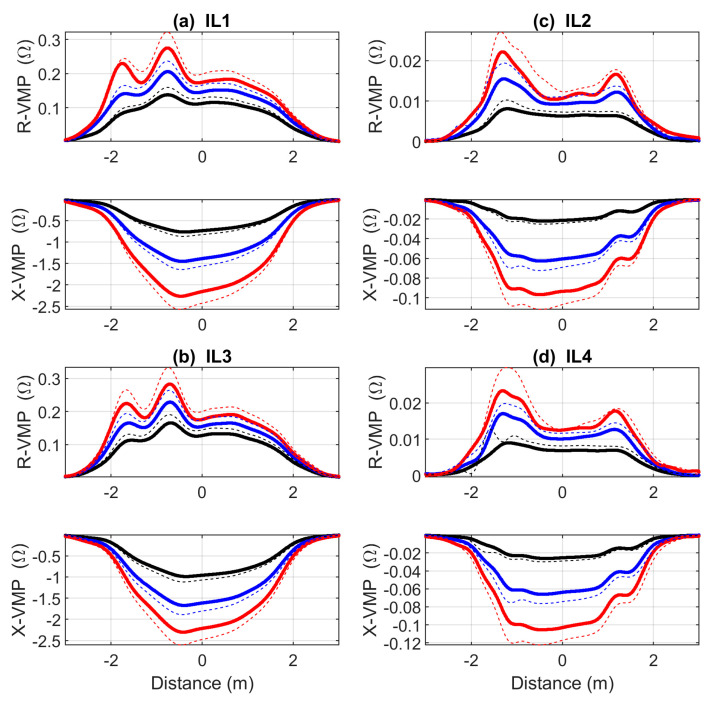
Exemplary VMPs of a Hyundai ix35 presented as a function of the distance traveled; thick lines apply to an unloaded car; thin dashed lines refer to a vehicle loaded with a mass of 306 kg; red VMPs are obtained at a high frequency component; blue VMPs are obtained at an intermediate value of the frequency component; black at the lowest frequency component, according to the operating frequencies listed in Table 1.

**Figure 4 sensors-23-02063-f004:**
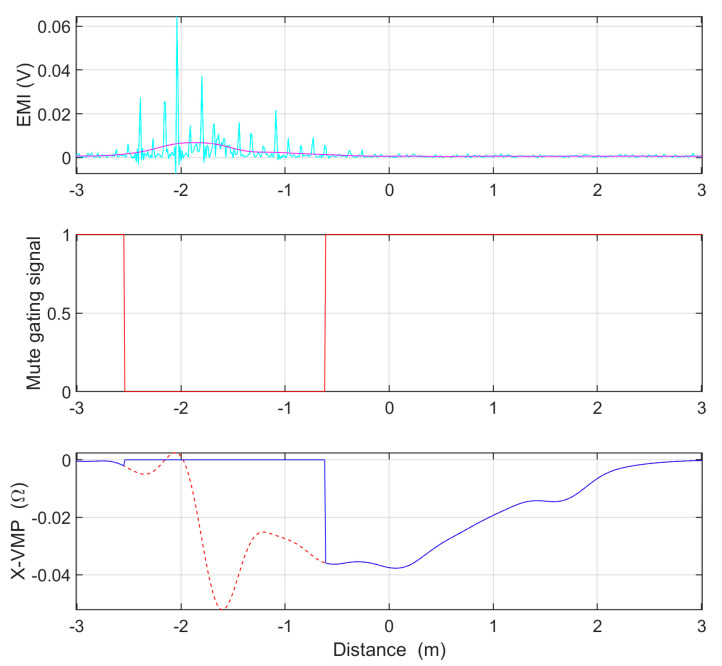
Example operation of the mute function. The oscillations in X-VMP (dashed red line) caused by EMI have been effectively muted.

**Figure 5 sensors-23-02063-f005:**
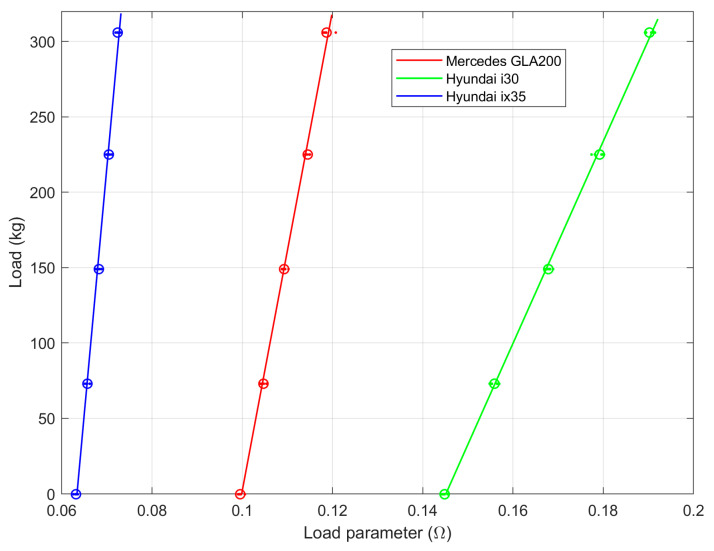
Dependencies between a car load and the load parameter for cars listed in the legend. Dots represent measurements; circles stand for mean values of the measured load parameter (for a set car and load); and lines show the least squares fit to the mean measurements (i.e., circles).

**Figure 6 sensors-23-02063-f006:**
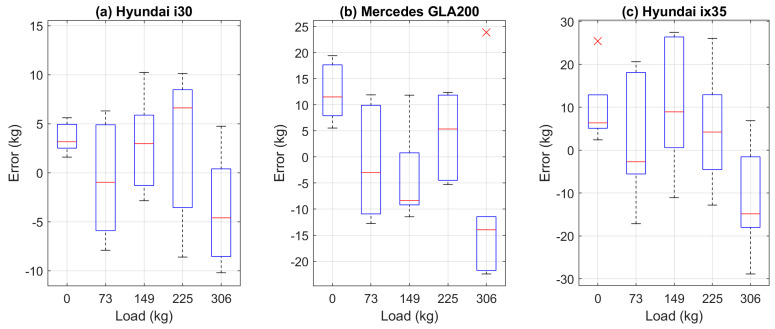
Measurement load error for test cars. On each box, the central red mark indicates the median, and the bottom and top edges of the box indicate the 25th and 75th percentiles, respectively. The whiskers extend to the most extreme data points not considered outliers, and the outliers are plotted individually using the ‘×’ symbol [28].

**Table 1 sensors-23-02063-t001:** The list of excitation frequencies applied in the system.

Frequency Value in kHz in a Given Channel:	f1	f2	f3
#1: for the first standard IL1 sensor	10	18	27
#3: for the second standard IL3 sensor	13	21	28
#2: for the first slim IL2 sensor	6	15	22
#4: for the second slim IL4 sensor	7	16	24

Where: f1, f2, f3—denote excitation frequencies.

**Table 2 sensors-23-02063-t002:** Technical data of the cars used in experiment and driver’s weight.

Car Model	Car Weight (kg)	Permissible Load (kg)	Driver Weight (kg)	M (%)
Mercedes GLA200	1320	600	85	65.1
Hyundai i30	1193	527	68	71.2
Hyundai ix35	1366	464	82	83.6

Where M = (Driver weight + 306)/(Permissible load) 100%.

**Table 3 sensors-23-02063-t003:** The scaling factors (y = **a**x + **b**) of load measurement system and sensitivities.

Car Model	a (kg/Ω)	b (kg)	S (Ω/kg)
Mercedes GLA200	16,063.551	−1609.901	62.25 × 10^−6^
Hyundai i30	6702.737	−972.612	149.2 × 10^−6^
Hyundai ix35	32,574.053	−2064.104	30.7 × 10^−6^

Where: S—sensitivity factor, S = 1/a.

## Data Availability

Not applicable.

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
