# Peer review of "Load Estimation of Moving Passenger Cars Using Inductive-Loop Technology"

_sensors, 2023, doi:10.3390/s23042063_

Round 1

Reviewer 1 Report

Reviewer's Notes

1. Introduction needs improvement:

1.1. Giving nine references to one sentence (line 38) requires a more detailed discussion of the research problem.

1.2. The second paragraph (lines 39-50) is a summary of the work and research (should be deleted).

1.3. Lines 59-62 are table of contents (should be removed).

1.4. The introduction should be extended with the justification of research on passenger vehicles, because in practice and in scientific research, the most important are the methods of measuring the load on trucks, which are important for traffic safety and road infrastructure facilities. They are extensively described on the pages of the MDPI Publishing House.

1.5. At the end of the introduction (as a paragraph), the purpose and scope of the research as well as research hypotheses (which can be referred to later in the discussion) should be clearly defined.

2. Chapter „Experiment”

2.1. The photos provided in Appendix B show that the measurements for the cars discussed in the results were made in November. Were tests carried out on a different date for these three cars (Mercedes, Hyundai i30 and ix35)? If so, why are they not shown? In the Discussion, the authors (in line 227) refer to research in different seasons of the year, which is not included in the methodology or in the results. Only presented, as separate studies in Appendix C, are the Volkswagen Polo August studies.

2.2. The driver is also a burden for the car (especially a passenger car, where it is a greater percentage of the total GVW weight) affecting the ground clearance (VMP). The omission of the driver's weight (they were different in the studies) requires better explanation and justification.  

2.3. 306 kg was adopted in the research as a maximum weight of the load (without a driver). What is the maximum, permissible load with persons and luggage of the studied models? To what extent do the adopted load in the research do these limits?

3. Chapter "Results" requires a better discussion of the results obtained (linking them with the presented appendices), so that they well implement the clearly defined research goal and hypotheses and provide the basis for a good discussion.

4. Chapter "Discussion" needs improvement:

4.1 The text in lines 200-208 is not a discussion, and the statements contained therein are not based on research.

4.2. There is no reference to the accuracy of the tests compared to other measurement methods from the literature.

4.3. The statements contained in lines 227-232 do not result from the presented research and results, and no reference is made to any literature.

4.4. Car overload tests mainly concern trucks, and in the discussion the authors do not refer to the need for such tests for trucks, supporting it with relevant literature.

Reviewer 2 Report

The work presents an extension of the author’s previous research. The novelties and improvements of the current investigations are clearly emphasized compared to the previous ones.

In the conclusion of the paper, all shortcomings and the need for further research are highlighted, therefore I recommend this paper for publication.

The only thing I would draw attention to is that although you refer to previous work in the paper, you should describe the principle of the work in a little more detail in the introductory part and define the abbreviations in the first place where they are referred to in the paper so that it is clear to future readers (as is the case with the abbreviation IL sensors).

Reviewer 3 Report

This work reports a method to estimate the load of moving passenger car, which shows potential application in road safety. But there are some specific issues, the authors should address them before acceptance.

(1) Please reorganize and rewrite the introduction section, it lacks of clear logic.

(2) How are the durability and precision of the slim inductive-loop sensor?

(3) Please compare the sensitivity of the slim inductive-loop sensor with other sensors for monitoring the load estimation.

Round 2

Reviewer 1 Report

The authors significantly improved and completed the text of the manuscript. The responses to the reviewer are satisfactory.

Reviewer 3 Report

This manuscript can be accepted in this state.